# Magnetosheath plasma flow model around Mercury

Daniel Schmid[1], Yasuhito Narita[1], Ferdinand Plaschke[1], Martin Volwerk[1], Rumi Nakamura[1], and Wolfgang Baumjohann[1]

[1]Space Research Institute, Austrian Academy of Sciences, Graz, Austria

**Correspondence:** Daniel Schmid, Space Research Institute, Austrian Academy of Sciences, Schmiedlstr. 6, 8042 Graz, Austria
(Daniel.Schmid@oeaw.ac.at)

**Abstract.**

The magnetosheath is defined as the plasma region between the bow shock, where the super-magnetosonic solar wind plasma is decelerated and heated, and the outer boundary of the intrinsic planetary magnetic field, the so-called magnetopause. Based on the Soucek-Escoubet magnetosheath flow model at the Earth, we present an analytical magnetosheath plasma flow model around Mercury. The model can be used to estimate the plasma flow magnitude and direction at any given point in the magnetosheath exclusively on the basis of the plasma parameters of the upstream solar wind. The model serves as a useful tool to trace the magnetosheath plasma along the streamline both in a forward sense (away from the shock) and a backward sense (toward the shock), offering the opportunity of studying the growth or damping rate of a particular wave mode or evolution of turbulence energy spectra along the streamline in view of upcoming arrival of BepiColombo at Mercury.

## 1 Introduction

The magnetosphere of a planet constitutes an obstacle to the super-magnetosonic solar wind. Upstream of the planet a bow shock emerges, because the interplanetary magnetic field (IMF), embedded in the solar wind, cannot simply penetrate the magnetosphere. At the bow shock the super-magnetosonic solar wind plasma is decelerated and heated. The region with the subsonic, heated plasma downstream of the bow shock is called magnetosheath. The magnetosheath plays an important role in the interaction between bow shock and magnetosphere as it conveys energy between the solar wind and the planetary magnetosphere.

One of the earliest magnetosheath plasma flow models is the hydrodynamic model introduced by *Spreiter et al.* (1966). Basically the model solves the gas-dynamic differential equations of an unmagnetized fluid around an obstacle, represented by the magnetosphere. It has successfully been tested against in-situ spacecraft data (*Song et al.*, 1999; *Stahara et al.*, 1993; *Spreiter and Alksne*, 1968) and applied to model the magnetospheres of various planets in our solar system (see *Stahara*, 2002, for a review). A decisive drawback of this model, however, is the high complexity and computational demands to calculate numerically a set of differential equations.

To reduce the computational complexity, several analytical plasma flow models have alternatively been proposed (*Russell et al.*, 1983; *Kallio and Koskinen*, 2000; *Romashets et al.*, 2008). An analytical magnetosheath flow model, which has success-

fully been tested against spacecraft observations at Earth, has been implemented by *Soucek and Escoubet* (2012). This model is based on the magnetic field model developed by *Kobel and Flückiger* (1994) and later modified and extended by *Génot et al.* (2011) to obtain a magnetosheath plasma flow model. The essential advantage of this model is its compatibility with a wide range of bow shock and magnetopause models while retaining the simplicity and computational efficiency of the original magnetic field model. Furthermore, the model allows to calculate the plasma flow velocity at any point in the magnetosheath using only the spacecraft position and solar wind parameter upstream of the bow shock.

In this work we follow the procedure proposed by *Soucek and Escoubet* (2012) and rescale their terrestrial magnetosheath flow model to the space environment at Mercury. First, we introduce the Hermean bow shock and magnetopause model, used to obtain the magnetosheath plasma flow model. Second, we revisit the magnetic field model of *Kobel and Flückiger* (1994) which *Soucek and Escoubet* (2012) used to determine the plasma velocity direction in the magnetosheath. Third, we extend the model by the Rankine-Hugoniot relations in a similar way as *Génot et al.* (2011) to determine the velocity magnitude downstream of the shock.

The aim of this paper is to provide a tool to estimate the plasma flow at a given point of spacecraft observation inside the Hermean magnetosheath on the basis of the solar wind conditions.

## 2   Bow shock and magnetopause model at Mercury

In the following we use an aberrated Mercury Solar Magnetospheric (MSM) coordinate system. This coordinate system is based on the Mercury Solar Orbital (MSO) coordinate system, but its origin is shifted northward by $479\,\mathrm{km}$ from the MSO origin to account for Mercury's dipole offset and rotated into the solar wind velocity direction. In the MSO coordinate system the $X_{\mathrm{MSO}}$-axis points sunward, the $Y_{\mathrm{MSO}}$-axis points anti-parallel to Mercury's orbital velocity, and $Z_{\mathrm{MSO}} = X_{\mathrm{MSO}} \times Y_{\mathrm{MSO}}$ completes the right-handed system. To compensate for the aberration of the solar wind direction due to the orbital motion of Mercury around the sun, the $X_{\mathrm{MSO}}$-axis is rotated anti-parallel to the solar wind flow velocity direction. In the MSM coordinate system the bow shock and magnetopause models are considered to be cylindrically symmetric around the $X_{\mathrm{MSM}}$-axis, reducing the three dimensions $\{X_{\mathrm{MSM}}, Y_{\mathrm{MSM}}, Z_{\mathrm{MSM}}\}$ to two dimensions $\{X_{\mathrm{MSM}}, \rho_{\mathrm{MSM}}\}$ with $\rho = \sqrt{Y_{\mathrm{MSM}}^2 + Z_{\mathrm{MSM}}^2}$.

*Slavin et al.* (2009) modeled the bow shock at Mercury by a conic section of the form

$$\xi = \sqrt{(x_{BS} - x_0)^2 + \rho_{BS}^2} = \frac{p\epsilon}{1 + \epsilon \cos\phi}, \tag{1}$$

with $x_0$ the distance of the focus of the conic section from the dipole center along $X_{\mathrm{MSM}}$, $\rho_{BS} = \sqrt{y_{BS}^2 + z_{BS}^2}$ the distance from the $X_{\mathrm{MSM}}$-axis, $p$ the focal parameter, and $\epsilon$ the eccentricity. With the advent of the MESSENGER (MErcury Surface, Space ENvironment, GEophysics and Ranging *Solomon et al.*, 2007) spacecraft in an orbit around Mercury, it was possible to characterize the spatial location of the bow shock and magnetopause statistically. *Winslow et al.* (2013) determined that the best-fit parameters to the bow shock are given by $x_0 = 0.5\,\mathrm{R_M}$, $p = 2.75\,\mathrm{R_M}$ and $\epsilon = 1.04$. With these parameters the extrapolated subsolar bow shock stand-off distance is $R_{\mathrm{BS}} = 1.9\,\mathrm{R_M}$ (Mercury radii, $1\,\mathrm{R_M} \sim 2440\,\mathrm{km}$). For this work it is

advantageous to transform Equation (1) into the origin of the MSM coordinate system with

$$
\begin{aligned}
r_{\mathrm{BS}} &= \sqrt{(\xi\cos\phi + x_0)^2 + (\xi\sin\phi)^2}, \\
\theta &= \arccos\left(\frac{\xi\cos\phi + x_0}{r_{\mathrm{BS}}}\right),
\end{aligned}
\tag{2}
$$

where $r_{BS}$ is the distance from the dipole center to the bow shock and $\theta$ the angle between $r_{\mathrm{BS}}$ and the $X_{\mathrm{MSM}}$-axis. Figure 1 shows a schematic illustration of the parameters $\xi$, $\phi$, $r_{\mathrm{BS}}$ and $\theta$, which are used in the formulation of the bow shock.

*Korth et al.* (2015) used the magnetopause model proposed from *Shue et al.* (1997) and found that the MESSENGER observations of magnetopause crossing are best-fit by

$$
r_{\mathrm{MP}} = \sqrt{x_{\mathrm{MP}}^2 + \rho_{\mathrm{MP}}^2} = R_{\mathrm{MP}}\left(\frac{2}{1 + \cos\theta}\right)^{\alpha},
\tag{3}
$$

with $\alpha = 0.5$ the best-fit flaring parameter, and $R_{\mathrm{MP}} = 1.42\,R_{\mathrm{M}}$ the subsolar stand-off magnetopause distance.

Figure 1 shows the the *Slavin et al.* (2009)-bow shock model (S09-BS) and *Korth et al.* (2015)-magnetopause model (K15-MP) evaluated from Equations (2) and (3), respectively.

## 3    The KF94 magnetic field model

To obtain the magnetosheath plasma flow direction we follow the procedure proposed by *Soucek and Escoubet* (2012) and use the magnetic field model developed by *Kobel and Flückiger* (1994). In the following we denote this model as KF94-model and mark all quantities pertaining to the KF94-model by a tilde ˜. In the KF94-model the bow shock (BS) and magnetopause (MP) at Mercury are modeled by parabolic surfaces at a common focus with

$$
\tilde{r}_{\{\mathrm{BS,MP}\}} = \frac{-\cos\theta + \sqrt{\cos^2\theta + 4R_{\{\mathrm{BS,MP}\}}b_{\{\mathrm{BS,MP}\}}\sin^2\theta}}{2b_{\{\mathrm{BS,MP}\}}\sin^2\theta},
\tag{4}
$$

with $b_{\mathrm{BS}} = 1/(4R_{\mathrm{BS}} - 2R_{\mathrm{MP}})$ and $b_{\mathrm{MP}} = 1/(2R_{\mathrm{MP}})$ defined by the stand-off distances $R_{\{\mathrm{BS,MP}\}}$.

Under the assumption that the IMF is parallel to the solar wind, the magnetic field lines of the KF94-model represent the flowlines of the solar wind and magnetosheath plasma. Using the magnetic field vector direction in the KF94-model, *Soucek and Escoubet* (2012) determined the flow velocity vector at a given position $\mathbf{r} = (x, \rho)$ by

$$
\begin{aligned}
\tilde{v}_{\mathrm{x}} &= v_{\mathrm{m}}\left(C/2d - C/R_{\mathrm{MP}}\right), \\
\tilde{v}_{\rho} &= v_{\mathrm{m}}\left(C\rho/[2d(d + x - R_{\mathrm{MP}}/2)]\right),
\end{aligned}
\tag{5}
$$

where $v_{\mathrm{m}}$ corresponds to the flow velocity magnitude, $d = |\mathbf{r} - \mathbf{r}_0|$ the difference between the given position in the magnetosheath and the parabolic surface focus $\mathbf{r}_0 = (R_{\mathrm{MP}}/2, 0)$, and $C = R_{\mathrm{MP}}(2R_{\mathrm{BS}} - R_{\mathrm{MP}})/(2R_{\mathrm{BS}} - 2R_{\mathrm{MP}})$ a constant defined by the bow shock and magnetopause stand-off distances.

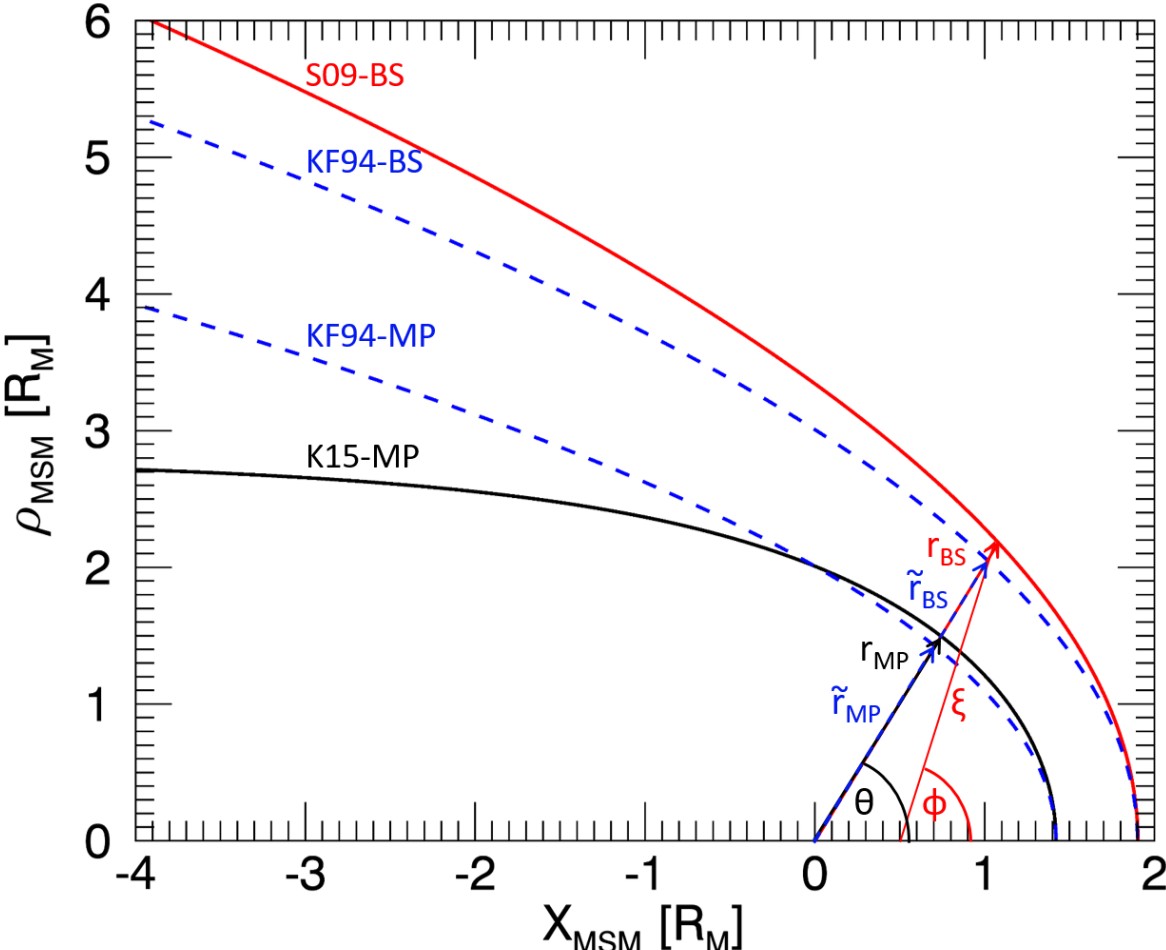

**Figure 1.** Schematic representation of the parameters used in the formulation of the bow shock and magnetopause in the MSM equatorial plane. The solid red line is the bow shock evaluated from Equation (2) (S09-BS, *Slavin et al.*, 2009). The solid black line represents the magnetopause determined by Equation (3) (K15-MP, *Korth et al.*, 2015). The dashed blue lines are the bow shock and magnetopause determined by Equation (4) from the KF94-model (*Kobel and Flückiger*, 1994).

## 4 The magnetosheath plasma flow model around Mercury

To obtain the magnetosheath plasma flow at a specific point ($\mathbf{r} = (x, \rho)$) in the magnetosheath at Mercury, we evaluate the plasma flow direction first, and then determine the magnitude of the velocity vector from the Rankine-Hugoniot relation across the bow shock.

A magnetic field model is used to describe the plasma flow here. The reason for this is explained as follows. Assumptions are made such that there are no proton sinks or sources in the magnetosheath. Strictly speaking, this assumption is only weakly

justified because of the neutral particles such as the hydrogen corona and the sodium exosphere around Mercury. A stationary case is taken in the fluid picture. The continuity equation reads then:

$$\nabla \cdot (n\mathbf{U}) = 0, \tag{6}$$

where $n$ and $\mathbf{U}$ are the density and velocity of protons, respectively. Now we make an analogy such that

$$(n\mathbf{U}) \longrightarrow \mathbf{B}, \tag{7}$$

holds, and Equation 6 is equivalent to the divergence-free condition of magnetic field,

$$\nabla \cdot \mathbf{B} = 0, \tag{8}$$

For a more detailed discussion see e.g. *Génot et al.* (2011).

## 4.1 Plasma flow direction

Following the procedure proposed by *Soucek and Escoubet* (2012), we rescale the plasma flow direction from the KF94-model to Mercury's space environment as follows:

1. As a first step we calculate the angle $\theta = \arccos(x/\sqrt{x^2 + \rho^2})$ between $\mathbf{r}$ and the $X_{\mathrm{MSM}}$-axis.

2. Then we estimate the fractional distance, $\mathcal{F}$, of $\mathbf{r}$ between the bow shock and magnetopause from Equations (2) and (3) with

$$\mathcal{F} = \frac{r(\theta) - r_{\mathrm{BS}}(\theta)}{r_{\mathrm{BS}}(\theta) - r_{\mathrm{MP}}(\theta)}. \tag{9}$$

3. Now we change into the KF94-model and calculate $\tilde{r}_{\mathrm{BS}}(\theta)$ and $\tilde{r}_{\mathrm{MP}}(\theta)$ from Equation (4) with the angle $\theta$. Note that the stand-off distances ($R_{\mathrm{BS}}$ and $R_{\mathrm{MP}}$) and the focus ($\mathbf{r}_0 = (R_{\mathrm{MP}}/2, 0)$) in Equation (4) are the best-fit values from Equations (2) and (3).

4. In a next step we determine the fractional position within the magnetosheath in the KF94-model with

$$\tilde{r}(\theta) = \mathcal{F}[\tilde{r}_{\mathrm{BS}}(\theta) - \tilde{r}_{\mathrm{MP}}(\theta)] + \tilde{r}_{\mathrm{BS}}(\theta), \tag{10}$$

according to Equation (9).

5. Using Equation (5) we evaluate the KF94 flow velocity vector, $\tilde{\mathbf{v}}$, at the position $\tilde{r}(\theta)$. Note that the velocity magnitude $v_{\mathrm{m}}$ is determined in a later step.

6. With the obtained flow velocity vector, $\tilde{\mathbf{v}}$, we are able to estimate the new position of an adjacent point along the same flowline $\tilde{\mathbf{r}}' = \tilde{\mathbf{r}} + \tilde{\mathbf{v}}\Delta t$, by choosing an infinitesimally small time increment $\Delta t$.

7. Next we determine the angle between the new position and the $X_{\mathrm{MSM}}$-axis, $\theta'$, and the fractional distance inside the KF94 magnetosheath, $\mathcal{F}'$, using Equation (9).

8. Finally we transform the new position $\tilde{\mathbf{r}}'(\theta')$ back from the KF94-model to the original MSM reference frame where the magnetosheath is confined by Equations (2) and (3). The new position, $\mathbf{r}'$, inside this magnetosheath is then given by

$$\mathbf{r}' = \mathcal{F}'\left[\mathbf{r}_{\mathrm{BS}}(\theta') - \mathbf{r}_{\mathrm{MP}}(\theta')\right] + \mathbf{r}_{\mathrm{BS}}(\theta'), \tag{11}$$

and thus the plasma flow direction can be determined by $\mathbf{v} = (\mathbf{r}' - \mathbf{r})/\Delta t$.

Applying recursively this procedure (step(1)-(8)) yields the plasma flowline within Mercury's magnetosheath. In Fig.2 five examples of flowlines are shown.

## 4.2 Plasma flow magnitude

To evaluate the magnetosheath plasma velocity magnitude, $v_{\mathrm{m}}$, we apply the Rankine-Hugoniot (RH) equations, which relate the upstream ($u$) with the downstream ($d$) plasma conditions. The downstream plasma flow velocity directly behind the bow shock, $v^d$, is derived by the following procedure:

1. From the given spacecraft position in the magnetosheath, $\mathbf{r} = (x, \rho)$, we trace the flowline back to the bow shock. Thereto we iteratively apply steps (1)-(8) from above, with reversed increments $\tilde{\mathbf{r}}' = \tilde{\mathbf{r}} - \tilde{\mathbf{v}}\Delta t$ in step (6), until the bow shock is reached ($\mathcal{F}' = 0$). Then we calculate the angle $\theta_{\mathrm{BS}}$ between the $X_{\mathrm{MSM}}$-axis and the bow shock intersection at $(x_{\mathrm{BS}}, \rho_{\mathrm{BS}})$ with $\theta_{\mathrm{BS}} = \arccos(x_{\mathrm{BS}}/\sqrt{x_{\mathrm{BS}}^2 + \rho_{\mathrm{BS}}^2})$.

2. In a next step we determine the bow shock tangent $\hat{\mathbf{t}}$ and normal $\hat{\mathbf{n}}$ unit vectors where the back-traced flowline intersects the bow shock. For any point along the bow shock, the normal, $\mathbf{n}$, and tangent, $\mathbf{t}$, vector can easily be computed by

$$
\begin{aligned}
\mathbf{t} &= \left[\frac{\mathrm{d}r_{\mathrm{BS}}}{\mathrm{d}\theta}\cos\theta - r_{\mathrm{BS}}\sin\theta\right]\hat{\mathbf{e}}_{\mathrm{x}} + \left[\frac{\mathrm{d}r_{\mathrm{BS}}}{\mathrm{d}\theta}\sin\theta + r_{\mathrm{BS}}\cos\theta\right]\hat{\mathbf{e}}_{\rho}, \\
\mathbf{n} &= \left[\frac{\mathrm{d}r_{\mathrm{BS}}}{\mathrm{d}\theta}\sin\theta + r_{\mathrm{BS}}\cos\theta\right]\hat{\mathbf{e}}_{\mathrm{x}} - \left[\frac{\mathrm{d}r_{\mathrm{BS}}}{\mathrm{d}\theta}\cos\theta + r_{\mathrm{BS}}\sin\theta\right]\hat{\mathbf{e}}_{\rho},
\end{aligned} \tag{12}
$$

where $\frac{\mathrm{d}r_{\mathrm{BS}}}{\mathrm{d}\theta}$ is numerically calculated from two consecutive points along the bow shock given by Equation (2). The shock reference frame of reference is then defined by the normalized normal and tangent vector at $\theta_{\mathrm{BS}}$.

3. In the shock reference frame the RH-equations can be combined to determine the downstream velocity vector component parallel $v_n^d$ and perpendicular $v_t^d$ to the shock normal (see e.g. *Génot*, 2008)

$$
\begin{aligned}
v_n^d &= v_n^u\frac{1}{\mathcal{R}}, \\
v_t^d &= v_n^u\left(\tan\theta_{\mathrm{Vn}} + \frac{1 - 1/\mathcal{R}}{\left[M_{\mathrm{A}}^{u\,2}/(\mathcal{R}\cos^2\theta_{\mathrm{Bn}})\right] - 1}\tan\theta_{\mathrm{Bn}}\right),
\end{aligned} \tag{13}
$$

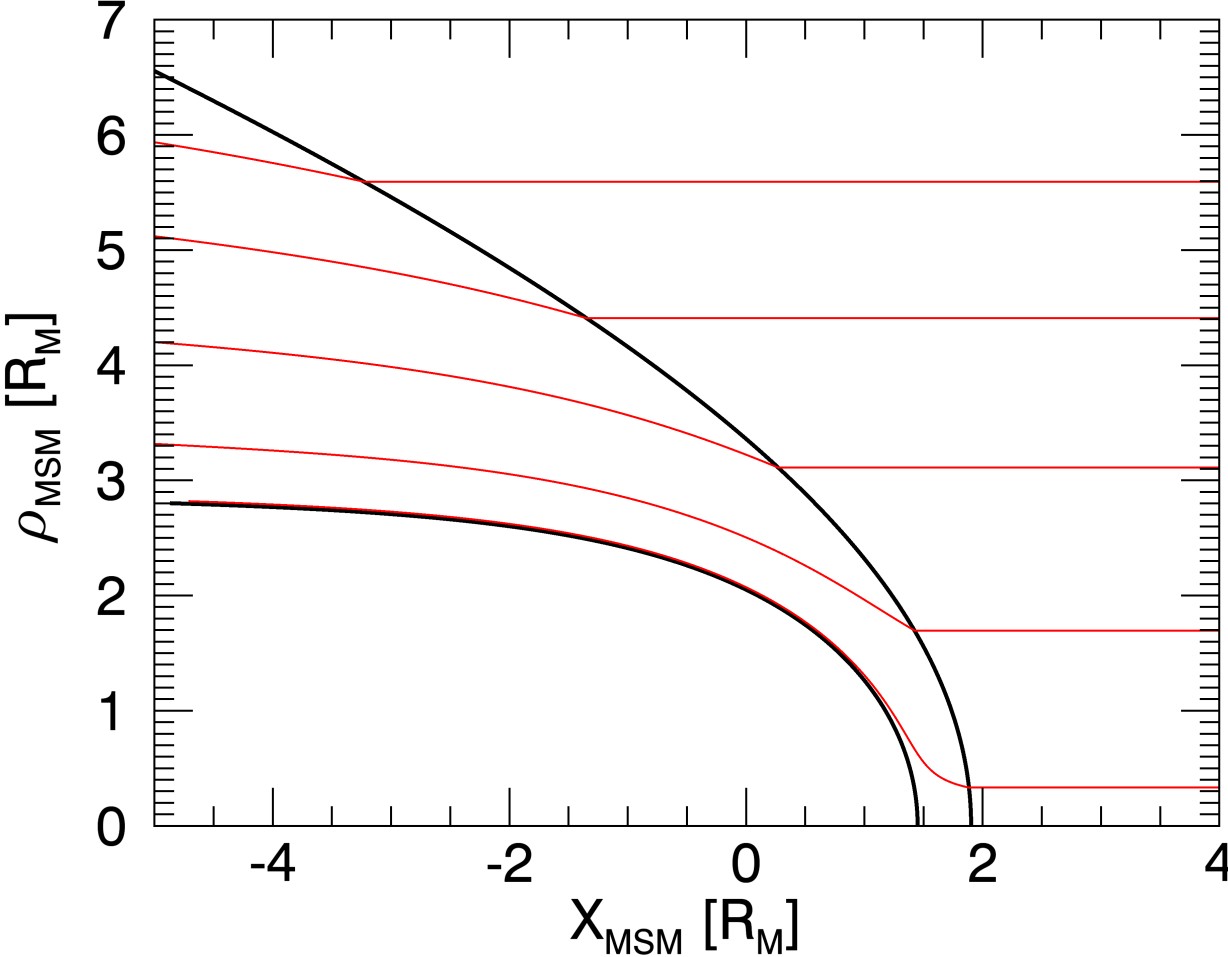

**Figure 2.** Schematic representation of the flowlines (red) in the MSM equatorial plane. The solid black lines are the bow shock and magnetopause evaluated from Equations (2) and (3), respectively.

where $v_n^u$ is the upstream velocity vector component parallel to the shock normal, $\mathcal{R} = \rho^d/\rho^u$ the compression ratio between the upstream and downstream mass density, $\theta_{\mathrm{Vn}} = \arctan(v_t^u/v_n^u)$ the angle between the upstream velocity vector and the shock normal, $\theta_{\mathrm{Bn}} = \arctan(B_t^u/B_n^u)$ the angle between upstream magnetic field vector and shock normal, and $M_{\mathrm{A}}^u = v_n^u \frac{\sqrt{\mu_0 \rho^u}}{B_n^u}$ the upstream Alfvén Mach number.

4. In Equation (13) all parameters pertain to the upstream side, except the compression ratio $\mathcal{R}$. However, $\mathcal{R}$ can also be expressed by exclusively upstream parameters with (see e.g. *Anderson*, 1963)

$$(M_A^{u\,2} - \mathcal{R})^2(\gamma\beta^u\mathcal{R} + M_A^{u\,2}\cos^2\theta_{\text{Bn}}[(\gamma-1)\mathcal{R}-(\gamma+1)])+$$
$$\mathcal{R}M_A^{u\,2}\sin^2\theta_{\text{Bn}}([\gamma+(2-\gamma)\mathcal{R}]M_A^{u\,2}+\mathcal{R}[(\gamma-1)\mathcal{R}-(\gamma+1)]) = 0 \tag{14}$$

where $\beta^u = (2\mu_0 p^u)/B^{u\,2}$ is the ratio of the upstream thermal to magnetic pressures, and $\gamma$ the polytropic index which is typically assumed to be $\gamma = 5/3$. Equation 14 is equivalent to Equation 2.43 in *Anderson* (1963) with different notations. The solution is given in the form of compression ratio as a function of the shock angle $\theta_{\text{Bn}}$. Solutions exist for a compression ratio in the range $1 \leq \mathcal{R} \leq 4$ when using Equation 14. Another class of solutions also exists for the expansion ($\mathcal{R} < 1$) with a decrease of entropy from the upstream onto the downstream side. The latter case is not physically relevant, and is not considered here. The upper limit of compression ratio ($\mathcal{R} = 4$) corresponds to the limit of high Alfvén Mach number ($M_A \to \infty$) under a polytropic index of $\gamma = 5/3$.

5. By solving Equation 14 for $\mathcal{R}$ the downstream velocity magnitude $v_{\text{d}} = \sqrt{v_n^{d\,2} + v_t^{d\,2}}$ is therefore entirely determined by only the upstream plasma parameters.

Since the detailed density profile along the flowline is unknown, we assume in a first approximation a constant plasma density and thus a constant velocity magnitude along the flowline. Therefore, the velocity magnitude at a given point $\mathbf{r}$, directly corresponds to the velocity magnitude downstream of the shock and $v_{\text{m}} = v_{\text{d}}$. Although this assumption has the tendency to underestimate the velocity close to the magnetopause, it yields satisfactory results in a first approach (*Génot et al.*, 2011).

The entire procedure from above is implemented in an IDL computer program which can be retrieved from OSF (*Schmid*, 2020). The program is designed to evaluate the plasma flow velocity vector at a given observation point of a spacecraft inside the Hermean magnetosheath exclusively on basis of the upstream solar wind conditions. As the solar wind input parameters we use the solar wind propagation model of *Tao et al.* (2005) which is modified by the orbital motion of Mercury. The model is one-dimensional magnetohydrodynamic, and takes the OMNI dataset as input to compute propagation at all solar system bodies including Mercury. The correction for the orbital motion is achieved by adding the solar wind velocity vector, $\mathbf{V}_{\text{SW}}$, (which is radially away from the Sun) and the orbital motion velocity vector of Mercury, $\mathbf{V}_{\female}$ with $\mathbf{V} = \mathbf{V}_{\text{SW}} + \mathbf{V}_{\female}$. To obtain $\mathbf{V}_{\female}$ we use the dataset provided from Navigation and Ancillary Information Facility (NAIF, *Acton*, 1996) which provide the distance between Mercury and Sun, $D$, and absolute velocity of Mercury, $V_{\female}$. To determine aberration velocity vector, we first calculate the aberration angle $\phi$ on basis of Mercury's elliptical orbit with

$$\phi = \arctan\left(b/\sqrt{a^2-b^2}\sin\left[\cos^{-1}((1-p/D)/e)\right]\right), \tag{15}$$

where $a$ is the semimajor axis, $b$ the semiminor axis, $p$ the semi-latus rectum and $e$ the eccentricity of the ellipse. With the aberration angle $\phi$ and the consideration whether Mercury moves towards/away from the sun we subsequently obtain the aberration velocity vector $\mathbf{V}_{\female,\text{x}} = \pm V_{\female}\sin(\phi)$ and $\mathbf{V}_{\female,\text{y}} = \pm V_{\female}\cos(\phi)$. The transformation due to this abberation effect is

made by applying a two-dimensional rotation matrix to the spatial coordinates (spanning the $x$ and $\rho$ coordinates):

$$175 \quad \begin{pmatrix} x^{'} \\ y^{'} \end{pmatrix} = \begin{pmatrix} \cos\theta_{\mathrm{a}} & -\sin\theta_{\mathrm{a}} \\ \sin\theta_{\mathrm{a}} & \cos\theta_{\mathrm{a}} \end{pmatrix} \begin{pmatrix} x \\ \rho \end{pmatrix}, \qquad (16)$$

where the aberration angle $\theta_{\mathrm{a}}$ is given by the radial solar wind velocity, $\mathbf{V}_{\mathrm{SW}}$, and the apparent solar wind velocity, $\mathbf{V}$ with $\theta_{\mathrm{a}} = \arccos\left(\mathbf{V}_{\mathrm{SW}} \cdot \mathbf{V}\right)$. In Fig.3 the results of the model are shown for the average solar wind plasma parameters during the entire MESSENGER operation service between 2011 and 2015. After modifying the solar wind velocity vector of the *Tao et al.* (2005) model by the orbital motion of Mercury (*Acton*, 1996), the average input solar wind plasma parameters for our model are: density of $n^u \approx 40\,\mathrm{cm}^{-3}$, temperature of $T \approx 18\,\mathrm{eV}$, flow speed of $|\mathbf{v}^u| \approx -400\,\mathrm{km/s}$, magnetic field magnitude of $|\mathbf{B}^u| \approx 20\,\mathrm{nT}$ with the radial component $B_{\mathrm{r}} \approx 18\,\mathrm{nT}$ (ignoring the sign) and the tangential component $B_{\mathrm{t}} \approx 16\,\mathrm{nT}$ (ignoring the sign). The mean values of density, temperature, flow speed, and magnetic field are valid for nearly 1500 days of observations of MESSENGER (confirmed by one of the reviewers). It is worthwhile to note that one finds an angle of $23°$ from the Tao dataset, which is consistent to the spiral angle at Mercury at an average position of $0.4$ astronomical units (AU) from the Sun for a solar wind speed of $400\,\mathrm{km/s}$. The $B_{\mathrm{x}}$ component is computed from the $B_{\mathrm{y}}$ component using the Parker spiral field in the Tao model. The Alfvén Mach number in the upstream region is $M_{\mathrm{A}} = v^u/V_{\mathrm{A}} \approx 5.8$ (with an Alfvén speed of $V_{\mathrm{A}} \approx 69\,\mathrm{km/s}$) and the plasma parameter beta (upstream) is $\beta = 2\mu_0 n k_{\mathrm{B}} T / B^2 \approx 0.72$ (where $\mu_0$ is the permeability of free space and $k_{\mathrm{B}}$ the Boltzmann constant) in our setup. Color coded is the obtained velocity magnitude $v_{\mathrm{m}}$. Additionally, the back-traced flowline from a virtual spacecraft located at $x_{\mathrm{MSM}} = -3\,\mathrm{R_M}$ and $\rho_{\mathrm{MSM}} = 3\,\mathrm{R_M}$ is plotted (green line). At the bow shock intersection the calculated shock normal $\hat{\mathbf{n}}$ and tangent $\hat{\mathbf{t}}$ are illustrated in thin black lines. The thick black line downstream of the bow shock shows the velocity vector direction evaluated from the RH relations, which is in good agreement with the streamline direction determined by the KF94-model. At the virtual spacecraft position the model predicts a magnetosheath plasma flow velocity of $v_{\mathrm{x}} \approx -200\,\mathrm{km/s}$ and $v_\rho \approx 17\,\mathrm{km/s}$.

A naive picture of the compression by a factor of $4$ (in the limit of high Mach number) is not realistic because the interplanetary magnetic field around Mercury reaches a magnitude of $20$ to $50\,\mathrm{nT}$, and the Alfvén Mach number is correspondingly smaller than that around the Earth by 20 to 50, respectively. A picture of the constant density and a reduced flow speed to $1/4$ of the solar wind speed at the magnetopause (along the streamline tangential to the magnetopause) is not valid, either, since the adiabatic expansion breaks down and the model is not applicable to the flow in the sub-solar region and at the magnetopause.

## 5 Discussion and Conclusions

Here we present the first analytical magnetosheath plasma flow model for the space environment around Mercury. The model is based on the magnetosheath model by *Soucek and Escoubet* (2012), which has successfully been tested against spacecraft observations at Earth. The proposed model is relatively simple to implement and provides the possibility to trace the flowlines inside the Hermean magnetosheath.

The model presented in this paper is generally robust and easy to implement for its analytic expression using upstream parameters. It can help to determine the local plasma conditions of a spacecraft in the magnetosheath exclusively on basis of the

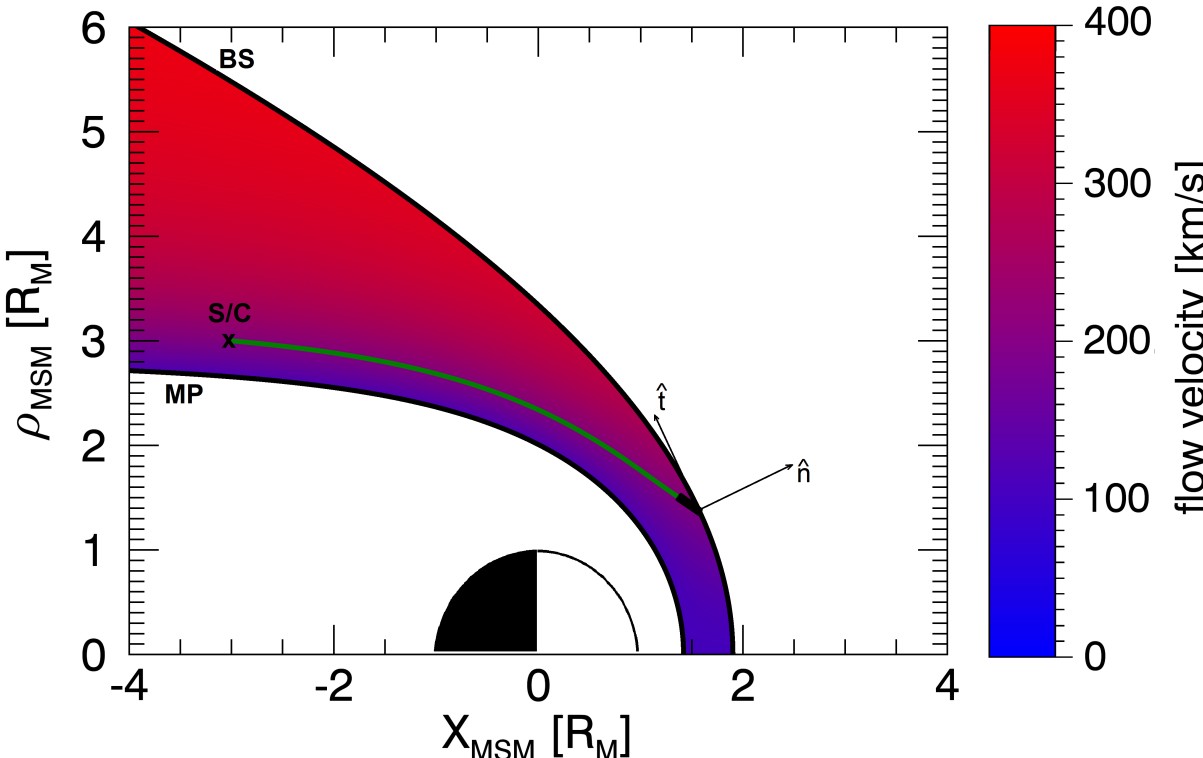

**Figure 3.** Color-coded flow speed calculated from Equation (13) with the averaged upstream parameters from the *Tao et al.* (2005) solar wind propagation model at Mercury between 2011 and 2015. Also plotted is a schematic representation of back-traced flowline (green) from a virtual spacecraft (S/C) to the bow shock (BS) with the respective shock normal $\hat{n}$ and tangent $\hat{t}$ obtained from Equation (12). The thick black line downstream of the BS is the downstream velocity vector direction determined by Equation (13). Alfvén Mach number in 5.8 and plasma beta is 0.79.

upstream solar wind parameters. Two applications are in mind in view of the BepiColombo mission, where the Mercury Magnetospheric Orbiter (MMO also referred to as Mio) will probe Mercury's magnetosheath and solar wind with unprecedented fast measurements of the particle distribution functions: (1) the Tátrallyay method to observationally determine the growth rate or damping rate of specific mode such as the mirror mode along the streamline (*Tátrallyay and Erdös*, 2002; *Tátrallyay et al.*, 2008), and (2) the Guicking method to observationally track the spectral evolution of turbulent fluctuations along the streamline (*Guicking et al.*, 2010, 2012).

At the moment a comparison with plasma data is technically not feasible for our model. Due to the limited particle measurements on board MESSENGER, it is not possible to obtain the plasma parameters properly in the solar wind and magnetosheath,

that is, by covering the full velocity distributions and to compare with the model velocities. Above all, the plasma instrument is located behind the heat shield and has just a limited field-of-view. Due to this fact, the majority (thermal core part) of solar wind particles cannot be detected. Comparison with the numerical simulations would be another possibility to test for the model, but a quantitative comparison remains a challenge for the reason that there are large discrepancies in the density and flow velocity among various simulation models on the dayside from the sub-solar point to the northern terminator (*Aizawa et al.*, 2021).

An approximation that the magnetic field is more aligned with the solar wind flow direction is more justified at Mercury than at the Earth because of the Parker spiral nature. One of the possible consequences of our assumption is that the magnetic field magnitude would change or evolve in the same sense as that of the plasma (or particle number flux) in the magnetosheath. The correlation between the magnetic field and that of the particle flux in the magnetosheath would ideally be tested against the plasma and magnetic field data on the arrival of BepiColombo at Mercury. The change in the number density can be interpreted as the change in the cross-sectional area of a fluxtube across which the plasma streams. The change in the flow velocity can then be compared with that from the adiabatic expansion and that from the measurement.

Following remarks are drawn as scientific message: First, Figure 1 visually demonstrates that different models predict different shapes of the tail and magnetosheath, which is an overlooked issue in the Mercury magnetosphere community. Second, Figure 2 shows that the flowlines near the sub-solar region (Sun-to-planet line if neglecting the planetary orbital motion) expand abruptly so that the adiabatic expansion may breaks down. In particular, the adiabatic expansion plays an important role in predicting the flow in the magnetosheath. Logical continuation of our model construction would be to evaluate also the density and velocity profile along the flowline and to test under which conditions the adiabatic expansion breaks down.

Although the proposed model has a good performance overall for a wide range of upstream conditions, the accuracy strongly depends on the used bow shock and magnetopause model. Here we utilize the bow shock and magnetopause model from *Slavin et al.* (2009) and *Korth et al.* (2015), which were adopted from MESSENGER boundary crossing observations.

The presented model is cylindrically symmetric around the $X_{\mathrm{MSM}}$-axis. In reality, however, non-radial IMF conditions will lead to a spatially asymmetric magnetosheath (*Nishino et al.*, 2008; *Dimmock and Nykyri*, 2013; *Dimmock et al.*, 2016). On the quasi-perpendicular side, where the shock-normal angles $\theta_{\mathrm{Bn}}$ are greater than $45°$, the magnetosheath is known to be thicker with larger plasma flow velocities than on the quasi-parallel side, where $\theta_{\mathrm{Bn}} < 45°$. Such asymmetries cannot be reproduced by the simple model presented here, but should be addressed in future work.

Furthermore, the method used to determine the flow velocity magnitude can possibly be improved. Here we assumed a constant plasma density and velocity along the flowline which has the tendency to underestimate the plasma velocity in regions with lower densities e.g. close to the magnetopause. *Génot et al.* (2011) proposed a simple ad-hoc model of a plasma density profile which has been implemented by *Soucek and Escoubet* (2012). While this ad-hoc density model showed good correspondence with in-situ spacecraft plasma observation at Earth, the solar wind and magnetospheric conditions at other planets can be very different (like e.g. at Mercury) and thus might give a worse prediction (*Soucek and Escoubet*, 2012). Our model inherits the properties from the Soucek-Escoubet model by scaling the Kobel-Flückinger model of the near-Earth environment: (1) time stationary flow, and (2) axially symmetric around the axis of (apparent) solar wind penetrating the planet. Assumption of time stationary flow may break down, when the change in the solar wind state is not negligible. Assumption of the axi-symmetric

magnetosheath may also break down, when the magnetopause location is not symmetric between the northern and the southern
hemisphere (in particular, in the tail region).

At this stage we decided not to include an ad-hoc density profile, also because it can hardly be tested due to the limited plasma observations around Mercury. The assumption of constant density implies a constant velocity along a given flow line. The velocity profile may vary considerably from that estimated in the earlier models such as the Spreiter model (*Spreiter et al.*, 1966), the Genot model (*Génot et al.*, 2011), and the Soucek-Escoubet model *Soucek and Escoubet* (2012). As mentioned above, the constant density will likely underestimate the propagation timing in the magnetosheath. In a future work such a density profile should be evaluated and included.

*Code availability.* An IDL program to evaluate plasma flow velocity vector in Mercury's magnetosheath from solar wind parameters of the Tao solar wind propagation model can be retrieved from OSF: https://osf.io/9jgqn/?view_only=2624aca3774c4ba8885dcb21a13e1b08 (*Schmid*, 2020).

*Data availability.* The plasma data of the heliospheric Tao model are open access and can be retrieved on the AMDA website (http://amda.cdpp.eu/, Centre de Données de la Physique des Plasmas (CDPP), 2018) via the WorkSpace Explorer: DataBase/Solar Wind Propagation Models/Tao Model/SW Input OMNI (*Tao et al.*, 2005). The orbital motion data of Mercury are provided by the Navigation and Ancillary Information Facility (NAIF) and can be retrieved on the NAIF website under https://wgc.jpl.nasa.gov:8443/webgeocalc (*Acton*, 1996).

*Author contributions.* DS initiated this study, collected the data and implemented the method. FP, YN, MV and WB helped evaluating the manuscript.

*Competing interests.* The authors declare that they have no conflict of interest.

*Acknowledgements.* This work is financially supported by the Austrian Research Promotion Agency (FFG) ASAP MERMAG-4 under contract 865967.

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
