# Peer review of "Magnetosheath plasma flow model around Mercury"

_Annales Geophysicae, 2021_

## Referee Comment (RC1)

This paper presents an analytical magnetosheath model for the Mercury environment. The model is built on existing bricks that were published in the literature for the Earth and were adapted for Mercury, notably the models for the bow shock and the magnetopause. The procedure is well described and the code (in IDL) is given in reference such that the user can easily adapt the model to any other particular use. I have a number of non-blocking comments below that should be addressed by the authors before publication. The paper deserves publication.

Discussions points
- Do the authors foresee specific applications for their model at Mercury ?
- I would like to see a longer discussion on the limitation of constant density in the magnetosheath. As this hypothesis implies a constant velocity along a given flow line, it produces (Figure 3) a very different velocity patterns that the previous models (Spreiter et al., 1966; Génot et al., 2011, Soucek & Escoubet, 2012) and will likely underestimate propagation timing in the magnetosheath.

Solar wind at Mercury
I don't find the explanation on how the solar wind parameters at Mercury are computed very well described. It could be specified that the model is 1D MHD and takes OMNI as input to compute propagation at all solar system objects including Mercury. Then I don't understand what are the average theta_Vn and theta_Bn ? It does not make sense as these values depend on the position on the shock (and on the shock model itself). A note could be added on the average Parker angle at Mercury. From the Tao dataset one finds 23° which is also the value for Mercury at an average position of 0.4UA and solar wind velocity of 400 km/s. Note that the Tao Bx value is computed from the model By value and an hypothesis of Parker field line which is consistent with the identical 23° values above. I confirm the average values given in the paper (n=40 cm-3, T=18 eV, v=400 km/s, b=20 nT) for the ~1500 days of observations of the Messenger mission. As |v^u| is a magnitude I would indicate 400 instead of -400 km/s. Please compute the corresponding M_A and beta values and indicate them on Figure 3.

Orbital motion
Please explain the procedure for the orbital motion correction briefly described at L147. The exact transformation should be given, perhaps in a short appendix.

Equation 11
I'm grateful to the authors for bringing my attention to Anderson, 1963 that I had overlooked. I checked Equation 11 of the paper that is equivalent to Equation 2.43 of Anderson et al., 1963 although the notations are different. However, I have a problem with its solution. I ran the piece of code given by the authors, and I also used an independent method (Cardan for $3^{rd}$ degree polynoms) in Python; both approaches give the same solution for the compression ratio as a function of theta_Bn and it is reproduced on the figure below for nominal solar wind conditions (Ma=12, beta=3). The curve tends towards zero for perpendicular shocks. Entropy considerations preclude compression ratio below 1. So I'm wondering if the equation, hence the model, always leads to a physical solution for all solar wind inputs.

[Figure]

IDL code available at the OSF page
I was able to run the code satisfactorily and trace a low line from a point in the magnetosheath (see below). The code is well commented. This may be outside of the scope of this review though, but it shows that the work complies with reproducibility principles.

[Figure]

Minor points
Figure 2 : please specify the parameters (M_A, beta, …) for which the figure is drawn

Figure 3 : same comment

L123: reference of frame → frame of reference

L130 : the upstream magnetic field ; the shock normal

L131 : why adding the last part "w.r.t. the shock normal" ? it is unnecessary here.

L138 : Equation 4 → Equation 11

L158 : a word is missing in front of the reference

---

## Author Comment (AC1)

**Reply to referees**

Manuscript "Magnetosheath plasma flow model around Mercury" by Schmid et al.

We thank the both reviewers for taking time to carefully read and raise comments and suggestions. All the comments are well justified and helpful. The quality of manuscript has improved a lot thanks to the reviewer comments. We give our reply to each of the review comments in the following in red and changes in the manuscript are marked in blue.

Reviewer 1

[1.1]
> This paper presents an analytical magnetosheath model for the Mercury environment.
> The model is built on existing bricks that were published in the literature for the Earth
> and were adapted for Mercury, notably the models for the bow shock and the
> magnetopause. The procedure is well described and the code (in IDL) is given in reference
> such that the user can easily adapt the model to any other particular use. I have a number
> of non-blocking comments below that should be addressed by the authors before
> publication. The paper deserves publication.
>
> Discussions points:
> Do the authors foresee specific applications for their model at Mercury ?

Yes, and we added the following text (page 10, line 205--212):

"Two applications are in mind in view of the BepiColombo mission, where the Mercury Magnetospheric Orbiter (MMO also referred to as Mio) will probe Mercury's magnetosheath and solar wind with unprecedented fast measurements of the particle distribution functions: (1) the Tátrallyay method to observationally determine the growth rate or damping rate of specific mode such as the mirror mode along the streamline (Tátrallyay and Erdös, 2002; Tátrallyay et al., 2008), and (2) the Guicking method to observationally track the spectral evolution of turbulent fluctuations along the streamline (Guicking et al., 2010, 2012).

[1.2]
> I would like to see a longer discussion on the limitation of constant density
> in the magnetosheath. As this hypothesis implies a constant velocity along a given
> flow line, it produces (Figure 3) a very different velocity patterns that the previous
> models (Spreiter et al., 1966; Génot et al., 2011, Soucek & Escoubet, 2012) and will
> likely underestimate propagation timing in the magnetosheath.

True and agreed. We added a paragraph (page 12, line 253--256):

"The assumption of constant density implies a constant velocity along a given flow line. The velocity profile may vary considerably from that estimated in the earlier models such as the Spreiter model (Spreiter et al., 1966), the Genot model (Génot et al., 2011), and the Soucek-Escoubet model

(Soucek and Escoubet, 2012). The constant density will likely underestimate the propagation timing in the magnetosheath."

[1.3]
> Solar wind at Mercury
> I don't find the explanation on how the solar wind parameters at Mercury
> are computed very well described. It could be specified that the model
> is 1D MHD and takes OMNI as input to compute propagation at all solar
> system objects including Mercury.

Done. A sentence was added in the paragraph (page 8, line 163--165):

"The model is one-dimensional magnetohydrodynamic, and takes the OMNI dataset as input to compute propagation at all solar system bodies including Mercury."

[1.4]
> Then I don't understand what are the average theta_Vn and theta_Bn ?
> It does not make sense as these values depend on the position on
> the shock (and on the shock model itself).

We use the average |Br| and |Bt| values (which are the radial and tangential components of IMF) from the Tao datatset. The angles theta_Vn and theta_Bn are calculated from the averaged magnetic field components,|Br| and |Bt|, with respect to the bow shock normal, stream-wise traced upstream to the shock for the virtual spacecraft at at x = - 3 R_m and rho= 3 R_m. To avoid confusion, we use the averaged magnetic field components, |Br|≈18nT and |Bt|≈6nT, in the revision.

We added the follwing sentences (page 9 line 180--182):

"the average input solar wind plasma parameters for our model are: density of $n^u \approx 40$ cm−3, temperature of $T \approx 18$ eV, flow speed of $|v^u| \approx 400$ km/s, magnetic field magnitude of $|B^u| \approx 20$ nT with the radial component $Br \approx 18$ nT (ignoring the sign) and the tangential component $Bt \approx 6$ nT (ignoring the sign)."

[1.5]
> A note could be added on the average Parker angle at Mercury. From
> the Tao dataset one finds 23° which is also the value for Mercury
> at an average position of 0.4UA and solar wind velocity of 400 km/s.
> Note that the Tao Bx value is computed from the model By value and
> an hypothesis of Parker field line which is consistent with the identical
> 23° values above.

Agreed. Done. Page 9, line 184--187.

"It is worthwhile to note that one finds an angle of 23^o from the Tao dataset, which is consistent to the spiral angle at Mercury at an average position of 0.4 astronomical units (AU) from the Sun for a solar wind speed of

400 km s^-1. The B_x component is computed from the B_y component using the Parker spiral field in the Tao model.

[1.6]
> I confirm the average values given in the paper (n=40 cm-3, T=18 eV,
> v=400 km/s, b=20 nT) for the ~1500 days of observations of
> the Messenger mission. As |v^u| is a magnitude I would indicate
> 400 instead of -400 km/s.

Thank you! Included to page 9, line 182--184:

"The mean values of density, temperture, flow speed, and magnetic field are valid for nearly 1500 days of observations of MESSENGER (confirmed by one of the reviewers)."

[1.7]
> Please compute the corresponding M_A and beta values and
> indicate them on Figure 3.

Done (Fig. 3 caption, page 10). We also mention the Alfvén Mach number and the plasma beta value in the main text. See page 9, line 187--189.

"The Alfvén Mach number in the upstream region is $M_A = v^u / V_A \approx 5.8$ (with an Alfvén speed of $V_A \approx 69$ km s^−1 ) and the plasma parameter beta (upstream) is $\beta = 2\mu_0 n k_B T / B^2 \approx 0.72$ (where $\mu_0$ is the permeability of free space and $k_B$ the Boltzmann constant) in our setup."

[1.8]
> Orbital motion
> Please explain the procedure for the orbital motion correction briefly
> described at L147. The exact transformation should be given, perhaps
> in a short appendix.

The transformation is explained (page 8, line 164 to page 9, line 177):

"The model is one-dimensional magnetohydrodynamic, and takes the OMNI dataset as input to compute propagation at all solar system bodies including Mercury. The correction for the orbital motion is achieved by adding the solar wind velocity vector, $V_{SW}$, (which is radially away from the Sun) and the orbital motion velocity vector of Mercury, V' with $V = V_{SW} + V'$. To obtain V' we use the dataset provided from Navigation and Ancillary Information Facility (NAIF, Acton, 1996) which provide the distance between Mercury and Sun, D, and absolute velocity of Mercury, V'. To determine aberration velocity vector, we first calculate the aberration angle φ on basis of Mercury's elliptical orbit with

$$\varphi = atan(b/(sqrt(a^2-b^2)*sin(acos((1-p/D)/e))),    (15)$$

where a is the semimajor axis, b the semiminor axis, p the semi-latus rectum and e the eccentricity of the ellipse. With the aberration angle φ and the consideration whether Mercury moves towards/away from the sun we subsequently obtain the abberation velocity vector V'_x = (+/-) | V'|*sin(φ) and V'_y = | V'|*cos(φ). The transformation due to this abberation effect is made by applying a two-dimensional rotation matrix to the spatial coordinates (spanning the x and ρ coordinates):

$$\begin{pmatrix} x' \\ y' \end{pmatrix} = \begin{pmatrix} x \ \cos\theta_a - \rho \ \sin\theta_a \\ x \ \sin\theta_a - \rho \ \sin\theta_a \end{pmatrix} \begin{pmatrix} x \\ \rho \end{pmatrix}$$

where the abberation angle is given by the radial solar wind velocity V_SW and the apparent solar wind velocity V with $\theta_a = \cos^{-1}(V\_SW \cdot V)$

[1.9]
> Equation 11
> I'm grateful to the authors for bringing my attention to Anderson, 1963
> that I had overlooked. I checked Equation 11 of the paper that is equivalent
> to Equation 2.43 of Anderson et al., 1963 although the notations are different.
> However, I have a problem with its solution. I ran the piece of code
> given by the authors, and I also used an independent method
> (Cardan for 3rd degree polynoms) in Python; both approaches give the
> same solution for the compression ratio as a function of theta_Bn and
> it is reproduced on the figure below for nominal solar wind
> conditions (Ma=12, beta=3). The curve tends towards zero for perpendicular shocks.
> Entropy considerations preclude compression ratio below 1.
> So I'm wondering if the equation, hence the model, always leads
> to a physical solution for all solar wind inputs.

Thank you for the elaborate check! We only consider solutions between 1 ≤ R ≤ 4, otherwise the program stops and the model does not provide a solution. Although the equation also yields a solution for R<1, it has not a physical meaning, as that would entail a decrease in entropy from the upstream to the downstream side and thus would violate the second law of thermodynamics, as the referee correctly mentioned. Note that the upper limit R->4 is given in the high Alfvenic Mach number limit (M_A \rightarrow \infty) under the assumption gamma = 5/3.

We addded the following text (page 8, line 148--153):

"Equation (15) is equivalent to Eq. 2.43 in Anderson (1963) with different notations. The solution is given in the form of compression ratio as a function of the shock angle θ_Bn. Solutions exist for a compression ratio in the range 1 ≤ R ≤ 4 when using Eq. (12). Another class of solutions also exists for the expansion (R < 1) with a decrease of entropy from the upstream onto the downstream side. The latter case is not physically relevant, and is not considered here. The upper limit of compression ratio (R ≤ 4) corresponds to the limit of high Alfvén Mach number (MA → ∞) under

a polytropic index of γ = 5/3."

[1.11]
> IDL code available at the OSF page
> I was able to run the code satisfactorily and trace a low line from
> a point in the magnetosheath (see below). The code is well commented.
> This may be outside of the scope of this review though, but it shows
> that the work complies with reproducibility principles.

Thank you!

[1.12]
> Minor points
> Figure 2 : please specify the parameters (M_A, beta, ...) for which the figure is drawn

In Figure 2 five examples of streamlines are shown. No M_A and beta values are necessary.

[1.13]
> Figure 3 : same comment

Done. See Fig. 3 caption (page 10).

[1.14]
> L123: reference of frame -> frame of reference

Done. Page 6, line 134.

[1.15]
> L130 : the upstream magnetic field ; the shock normal

Done. Page 7, line 141.

[1.16]
> L131 : why adding the last part "w.r.t. the shock normal" ? it is unnecessary here.

Deleted. The sentence closes now with "the upstream Alfv\'en
Mach number." Page 7, line 142.

[1.17]
> L138 : Equation 4 -> Equation 11

Corrected to Eq. (14). Page 8, line 154.

[1.18]
> L158 : a word is missing in front of the reference

[revised manuscript text omitted]

---

## Author Comment (AC2)

==================================================

Reviewer 2

[2.1]
> The manuscript "Magnetosheath plasma flow model around Mercury"
> by Schmid et al., describes an interesting analytical plasma flow model
> which goal is to provide a computationally fast way to derive properties
> of the solar wind plasma in the Hermean magnetosheath.
> As the authors wrote, an analytical model, which would quickly provide
> estimations for plasma parameters in Mercury's magnetosheath, would be
> useful tool to interpret observations, for example, from the forthcoming
> BepiColombo Mercury mission.
> However, although the goal of the work is interesting and an analytical
> model may have potential to be used in studies where self-consistent
> 3D plasma models would be computationally too extensive, more details
> about the approach and validations of the model results are needed
> before the work is ready for publishing.
>
> Please see below my remarks and suggestions of how the manuscript
> could be improved.
>
> ------------------------------------------------------------------
>
> The authors describe in very detail different steps which are needed
> to derive the velocity field from a magnetic field model.
> That is very important issue, but at the moment, the manuscript
> looks to be a valuable technical documentation of the developed
> software tool, but not yet a comprehensive detailed research report.

We believe that it is a misunderstanding of the reviewer. It is true that
the manuscript has a technical aspect, but our manuscript offers
a theoretical model based on physics, and the goal of the manuscript
is well within the scope of AnGeo as scientific publication. We do have
scientific messages to the audience. First, Figure 1 visually demonstrates
that different models predict different shapes of the tail and magnetosheath,
which is an overlooked issue in the Mercury magnetosphere community and
the manuscript has a clear demonstration on that. Second, Figure 2 shows
that the flowlines near the sub-solar region (Sun-to-planet line if neglecting
the planetary orbital motion) expand abruptly so that the adiabatic expansion
may breaks down. In particular, the adiabatic expansion plays an important
role in predicting the flow in the magnetosheath. We address the both issues
in the last section as message to the audience. Ideally, we do plan to
continue the work but incorporation of the data is beyond the scope of the
current manuscript. Conditions for statistics must be tuned, for example, and
the manuscript volume can easily be inflated with too many goals.

We added the following paragraph (page 11, line 227--232):

" First, Figure 2 visually demonstrates that different models predict different shapes of the tail and magnetosheath, which is an overlooked issue in the Mercury magnetosphere community. Second, Figure 3 shows that the flowlines near the sub-solar region (Sun-to-planet line if neglecting the planetary orbital motion) expand abruptly so that the adiabatic expansion may breaks down. In particular, the adiabatic expansion plays an important role in predicting the flow in the magnetosheath. Logical continuation of our model construction would be to evaluate also the density and velocity profile along the flowline and to test under which conditions the adiabatic expansion breaks down."

[2.2]
> As an example, the authors do not describe in detail the basic physical
> assumptions to which the used equations are based on, but they rather
> refer to earlies works where the adopted original approaches are
> presented. For example, the reader may not easily understand why
> a magnetic field model is used to describe the plasma flow, because
> the manuscript does not mention that if there are no protons sinks
> and sources in the magnetosheath (which is not strictly speaking
> true because of the planetary hydrogen corona), in a fluid model
> stationary case the continuity equation is
>
>    $\nabla \cdot (n\,U) = 0,$    [Eq. 1]
>
> where n and U are the density and the velocity of protons, respectively.
> If we the make the analogy
>
>    $n\,U \leftrightarrow B,$      [Eq. 2]
>
> then a magnetic field (B) model would fulfil Eq. [1], because
> the magnetic field model has to always be divergence-free:
>
>    $\nabla \cdot B = 0,$    [Eq. 3]
>
> as discussed, for example, in Genot et al., 2011. Moreover,
> the relatively limited discussion of the model's theoretical background
> does not  provide an interested reader essential information to evaluate
> what consequence for the velocity field would be according to Eq. [1],
> if a non-constant plasma density (n) model are later used.

We accept the criticism. The comment by the reviewer is wonderful, and we added the explanation in the beginning of section 4 to remind the readers of the continuity equation and divergence-free magnetic field.

We added the following text (page 4, line 86 to page 5, line 95).

"A magnetic field model is used to describe the plasma flow here. The reason

for this is explained as follows. Assumptions are made such that there are
no proton sinks or sources in the magnetosheath. Strictly speaking,
this assumption is only weakly justified because of the neutral particles such
as the hydrogen corona and the sodium exosphere around Mercury. A stationary
case is taken in the fluid picture. The continuity equation reads then:

$$\nabla \cdot (nU) = 0,$$

where n and U are the density and the velocity of protons, respectively.
Now we make an analogy such that

$$nU \rightarrow B$$

holds, and then Eq. (6) is equivalent to the divergence-free condition of magnetic field,

$$\nabla \cdot B = 0.$$

For a more detailed discussion see e.g. Génot et al. (2011)."

[2.3]
> At the level where the manuscript is now, it does not have much new
> elements or inventions compared with the previous works made
> by Genot et al., 2011 and Soucek and Escoubet, 2012. In fact, the only
> new aspect looks to be that the authors have adopted models, which
> describe the shape of the Hermean bow shock and magnetopause.
> Implementing those boundaries is, or course, a mandatory 1st step
> to develop an analytical magnetosheath model for Mercury. However,
> an interested reader would appreciate more comparisons, especially
> quantitative comparisons, between the properties of the developed
> model and the properties of the works to which the used approach
> is based on (Genot et al., 2011, and Soucek and Escoubet, 2012),
> maybe also to Spretier&Stahara's gas dynamic model, to in situ
> observations and probably also, at least at some level, to self-consistent
> models (MHD or hybrid).

We accept the reviewer's comment (or spirit) that the model be
compared with the in situ data. This is unfortunately not feasible for
engineering reasons. Proper observations of particles around Mecury
are very difficult in space engineering. Due to the limited particle
measurements (limitation in angular, time, and energy resolution of
the elecrostatic analyzer) on board MESSENGER, it is not possible to
obtain the plasma parameters propery, that is, by covering the full
veloscity distributions, and to compare with the modeled. Above all,
the plasma instrument is located behind the heat shield and has just
a limited field-of-view. Due to this fact, the majority (thermal core part)
of solar wind particles cannnot be detected at all.

We, too, agree with the idea to compare some hybrid/MHD.
However, a quantitative comparison is again technically not feasible

for the reason that there are large discrepancies in the density and
flow velocity among various simulation models on the dayside
(from the subsolar point to the northern terminator).
Aizawa et al. (PSS, 2021) are reporting this discrepancy problem
by comparing different hybrid/MHD models at Mercury in detail.
Comparison with the model byGenot et al. (2011) and that by
Soucek and Escoubet (2012) is not benifical, since our model is
already based on the same assumptions and thus has the same
properties. Furthermore, a gas-dynamic data like e.g. Spreiter
and Sahara is not readily available.

We added a paragraph in conclusion section (page 11, line 213--219):

"Comparison with the MESSENGER plasma data is technically not feasible
for our model. Due to the limited particle measurements on board MESSENGER,
it is not possible to obtain the plasma parameters propery in the solar wind and
magnetoheath, that is, by covering the full veloscity distributions and to compare
with the model velocities. Above all, the plasma instrument is located behind the
heat shield and has just a limited field-of-view. Due to this fact, the majority
(thermal core part) of solar wind particles cannnot be detected. Comparison with
the numerical simulations would be another possibility to test for the model,
but a quantitative comparison remains a challenge for the reason that there are
large discrepancies in the density and flow velocity among various simulation
models on the dayside from the subsolar point to the northern terminator
(Aizawa et al., 2021)."

[2.4]
> Especially, in contrast to the work of Genot et al., 2011 and Soucek and
> Escoubet, 2012, the manuscript does not include clear validation of the
> developed model to the plasma observations made by the Messenger
> mission. It is, therefore, unclear how realistic, and therefore how useful,
> the developed model in practice is.

See the reply above (2.3).

[2.5]
> In fact, the authors express this concern also by themselves ("… Earth,
> the solar wind and magnetospheric conditions at other planets can be
> very different (like e.g. at Mercury) …"), but the manuscript does not
> provide much evidence of the realism of the developed Hermean
> magnetosheath model. For example, a reader may wonder how the
> approach where the space region which is in the Earth magnetosheath
> model of Kobel and Flückiger (1994), has been "stretched" and "fitted"
> inside to the space region which describes the presented Hermean
> magnetosheath model, affects the properties of the stream lines and,
> after including assumption to the plasma density, the speed of the solar wind.

We added the following sentences (page 11, line 246 to page 12, line 251).

"Our model inherits the properties from the Soucek-Escoubet model by scaling the Kobel-Fl\"uckinger model of the near-Earth environment: (1) time stationary flow, and (2) axially symmetric around the axis of (apparent) solar wind penetrating the planet. Assumption of time-stationary flow may break down, when the change in the solar wind state is not negligible. Assumption of the axi-symmetric magnetosheath may also break down, when the magnetopause location is not symmetric between the northern and the southern hemisphere (in particular, in the tail region)."

[2.6]
> As already mentioned, a more detailed discussion about the basic
> properties of the developed model is needed. For example, if it is assumed
> that the speed is constant along a stream line, then, because a flowline
> at the subsolar point forms the surface of the magnetopause, the
> density at the magnetopause is (for a high Mach values) about
> 4 times of the density in the undisturbed solar wind, and the bulk
> velocity in that situation would be about ¼ times the speed of the
> solar wind. How realistic those values are?

Very interesting observation! The answer is no; the compression factor does not reach 4 because of the low Mach number; the picture at the magnetopause with a constant density and a flow speed at 1/4 of the solar wind speed is not valid because the adiabatic condition breaks down in the subsolar region.

We added the following text (page 9, line 195--199):

"A naive picture of the compression by a factor of 4 (in the limit of high Mach number) is not realistic because the interplanetary magnetic field around Mercury reaches a magnitude of 20 to 50 nT, and the Alfvén Mach number is correspondingly smaller than that around the Earth by 20 to 50%, respectively. A picture of the constant density and a reduced flow speed to 1/4 of the solar wind speed at the magnetopause (along the streamline tangential to the magnetopause) is not valid, either, since the adiabatic expansion breaks down and the model is not applicable to the flow in the subsolar region and at the magnetopause."

[2.7]
> One could, for example, derive the velocity and the plasma density
> along some orbits of the Messenger spacecraft in order to illustrate
> how the properties of the plasma change in the magnetosheath and,
> if possible, compare predicted values with observations.

This will be done using MPPE (Mercruy Plasma Particle Experiment) data onboard the Magnetospheric Orbiter (MMO/Mio) on the arrival of BepiColombo at Mercury.

[2.8]
> It would also be very useful to analyse what consequences the derived
> velocity field would have to other plasma and field parameters.

> An approximation that the magnetic field is along the solar wind velocity
> vector is more justified at Mercury than at the Earth and, therefore,
> derivation of the magnetic field by assuming that the IMF is along
> the solar wind flow might be justified as a first order approximation.
> Would that mean (because of Eqs. [1] and [3] above) that the strength
> of the magnetic field would change similarly as the particle flux?
> If that is the case, it would be interesting, if possible, to see the value
> of the total magnetic field in the magnetosheath.

Very interesting question! Unfortunately, we do not have a clear answer
at the present stage of writing because of the lack of the spacecraft data.
But the reviewer's question is well justified and we address the question
as one of the possible tests once BepiColombo delivers the data after 2025.

We added the following text (page 11, line 220--226).

"An approximation that the magnetic field is more aligned with the solar wind
flow direction is more justified at Mercury than at the Earth because of the
Parker spiral nature. One of the possible consequences of our assumption is
that the magnetic field magnitude would change or evolve in the same sense
as that of the plasma (or particle number flux) in the magnetosheath. The
correlation between the magnetic field and that of the particle flux in the
magnethsheath would ideally be tested against the plasma and magnetic field
data on the arrival of BepiColombo at Mercury. The change in the number density
can be interpreted as the change in the cross-sectional area of a fluxtube across
which the plasma streams. The change in the flow velocity can then be compared
with that from the adiabatic expansion and that from the measurement."

[2.10]
> To summarize, as can also be seen in the comments above, the manuscript
> considers interesting and important plasma physical questions.
> The goal of the presented work is scientifically solid. However, as already
> mentioned, an interested reader would expect that the manuscript
> contains more detailed analysis of the basis of the approach and
> to see validations of the results.

We do wish to conduct the validation task, but a solid validation
task is not feasible at the present stage for the reasons explained
in the reply (2.3). We added more physical and scientific aspects in the revision
in the revision (thanks to the reviewer's comments) so that the manuscript
offers not only a model of magnetosheath but also lessons from
our model construction, e.g., use and breakdown of adiabatic expansion,
possible test methods, predictions.

[2.11]
> Although the results of such validation tests are at the moment
> unclear, based on the validations made of the models to which the
> presented model is based on, such validations have a good possibility
> to confirm the applicability of the developed model to provide

> a valuable description of the properties of the solar wind plasma
> in the Hermean magnetosheath.

We agree. To be done when the BepiColombo delivers the data.

[revised manuscript text omitted]

---

## Author Response (AR2)

**Reply to referees**

Manuscript "Magnetosheath plasma flow model around Mercury" by Schmid et al.

Again we would like to thank the both reviewers for taking time to review this manuscript and for the throughout very helpful suggestion to improve the quality. We give our reply to each of the review comments in the following in red and changes in the manuscript are marked in blue.

Editor

> I have a comment on the used symbolism. When the authors write cos^-1, I think that they
> may want to refer to arccos. This is not correct since cos^-1 means 1/cos.
> could you please check and correct this
True. We used the notation of an inverse function: cos^-1(x) and not the reciprocal trigonometry function (cos(x))^1.
To avoid any confusion we follow the Editor's suggestion and changed the notation accordingly to arccos, arcsin and arctan throughout the manuscript.

========================================================

Reviewer 1

> The authors produced a new version of the paper taking into account comments and
> remarks from both referees. In particular the second referee had some valuable criticisms
> on the physics and motivation of the model that have been adequately answered and
> improved the quality of the paper. The present paper is of publishable quality.

[1.1]
> I have one small comment on Equation 16 which holds 1 'cos' and 3 'sin' which looks a bit
> awkward for a rotation matrix; but I did not check precisely and this may be ok.
True. Equation 16 had two typos, which have both been corrected. Equation 16 now reads (Line 175 on page 9)

$$\begin{pmatrix} x' \\ y' \end{pmatrix} = \begin{pmatrix} \cos\theta_a & -\sin\theta_a \\ \sin\theta_a & \cos\theta_a \end{pmatrix} \begin{pmatrix} x \\ \rho \end{pmatrix}$$

[1.2]
> I note that the DOI for the 2 papers by Guicking are missing the 10.5194 prefix. These
> references in the text are not in italic. Like Aizawa et al. which is 2021 not 2011.
Done. The citations have been double-checked and corrected. Line 275 and 285-291, page 13.

========================================================

Reviewer 2

> The authors have carefully considered all comments, remarks and suggestions made by
>  the referee.

> The work is ready for publishing after correction of these three minor issues:

[2.1]
> 1) [Eq. 6, line 90] "nabla dot (nU)," -> "nabla dot (nU) = 0,"
Done. Equation 6 has been changed accordingly. Line 90, page 5

[2.2]
> 2) [Eq. 16, line 175]

>x cos(theta_a) -> cos(theta_a)
> x sin(theta_a) -> sin(theta_a)
> rho cos(theta_a) -> cos(theta_a)
> rho sin(theta_a) -> sin(theta_a)
Done. See comment [1.1]

[2.3]
> 3) [line 223] "… of the particle flux in the magnethsheath would …" -> "… of the particle flux
> in the magnetosheath would …"
Done. Typo has been corrected. Line 222, page 11.